# Cell-Free Filtrates (CFF) as Vectors of a Transmissible Pathologic Tissue Memory Code: A Hypothetical and Narrative Review

**DOI:** 10.3390/ijms231911575

**Published:** 2022-09-30

**Authors:** Jorge Berlanga-Acosta, Maday Fernandez-Mayola, Yssel Mendoza-Mari, Ariana Garcia-Ojalvo, Indira Martinez-Jimenez, Nadia Rodriguez-Rodriguez, Diana Garcia del Barco Herrera, Gerardo Guillén-Nieto

**Affiliations:** Tissue Repair, Wound Healing and Cytoprotection Research Group, Biomedical Research Direction, Center for Genetic Engineering and Biotechnology, Ave. 31 S/N. e/ 158 and 190, Cubanacán, Playa, Havana 10600, Cuba

**Keywords:** metabolic memory, diabetes, microangiopathy, arteriosclerosis, vascular remodeling, cellular memory, cancer, epigenetics

## Abstract

Cellular memory is a controversial concept representing the ability of cells to “write and memorize” stressful experiences via epigenetic operators. The progressive course of chronic, non-communicable diseases such as type 2 diabetes mellitus, cancer, and arteriosclerosis, is likely driven through an abnormal epigenetic reprogramming, fostering the hypothesis of a cellular pathologic memory. Accordingly, cultured diabetic and cancer patient-derived cells recall behavioral traits as when in the donor’s organism irrespective to culture time and conditions. Here, we analyze the data of studies conducted by our group and led by a cascade of hypothesis, in which we aimed to validate the hypothetical existence and transmissibility of a cellular pathologic memory in diabetes, arteriosclerotic peripheral arterial disease, and cancer. These experiments were based on the administration to otherwise healthy animals of cell-free filtrates prepared from human pathologic tissue samples representative of each disease condition. The administration of each pathologic tissue homogenate consistently induced the faithful recapitulation of: (1) Diabetic archetypical changes in cutaneous arterioles and nerves. (2) Non-thrombotic arteriosclerotic thickening, collagenous arterial encroachment, aberrant angiogenesis, and vascular remodeling. (3) Pre-malignant and malignant epithelial and mesenchymal tumors in different organs; all evocative of the donor’s tissue histopathology and with no barriers for interspecies transmission. We hypothesize that homogenates contain pathologic tissue memory codes represented in soluble drivers that “infiltrate” host’s animal cells, and ultimately impose their phenotypic signatures. The identification and validation of the actors in behind may pave the way for future therapies.

## 1. Introduction

Seminal evidence from more than 50 years ago showed that the ability to “learn and memorize” external signals is not an exclusive function of the central nervous system cells. The 1960s first saw experiments exemplifying that invertebrate organisms such as planarians are able to learn a task and transmit these memories to regenerating descendants following decapitation [1,2]. It was likely the prime evidence on the existence of what it is known today as “cellular memory”, an ancestral and evolutionarily conserved ability of non-neuronal cells to retain and use information [3,4,5]. This faculty allows the cell to “learn and recall” the experience of a primary insult, and assist in rapid cellular adaptation and survival if the same stressor subsequently appears [6].

The process of irreversible epigenetic imprinting on embryo stem cells to safeguard a lifetime differentiation program [7,8], and the ability of immune cells to remember a broad repertoire of antigenic challenges, are perhaps the most renowned examples of memory by non-neuronal cells in vertebrate organisms [9]. Other forms of cellular memory have been successively discovered in recent years: (1) fibroblasts’ topographic differentiation memory [10], (2) muscle cells mass memory [11,12], (3) skin inflammation memory [13], and (4) respiratory allergy inflammatory memory [14].

These fascinating events are founded and directed by a particular epigenetic code that in a simplified manner appears operated through three major processes: DNA/RNA methylation, post-translational modifications of histone proteins, and non-coding RNA modifications [15,16]. The long time preservation of a certain epigenetic writing, translates in the cellular ability to retain phenotypic and behavioral traits, typical of the donor organism from which they originated. This concept explains, for instance, why cultured cells irrespective to the number of passages and culturing conditions, preserve representative qualities of the donors’ pathology, as it is the case for fibroblasts and endothelial cells derived from diabetic subjects [17,18,19,20,21].

Diabetes, arteriosclerosis, and cancer are among the most prevalent chronic non-communicable diseases (NCD), representing leading causes of global morbidity and mortality [22]. The fact that these NCD are characterized by an undefined etiology, a long, silent and insidious latency period, and lack of definitive cure, explains the escalating number of associated deaths [23,24]. Epigenetic studies focusing diabetes, arteriosclerosis, and cancer; reveal the crucial role of a de novo established, abnormal, epigenetic reprogramming, originating a sort of “pathologic cellular memory”, implicated in the onset of a clinical phenotype, disease complications progression, and an individual’s clinical course [25,26,27].

Diabetes is likely the most convincing scenario to illustrate how a trivial exogenous factor such as the level of blood glucose may reshape the epigenetic landscape and ultimately build a pathologic memory [28,29]. The existence of this hyperglycemia-mediated metabolic stress memory explains why prior hyperglycemic exposure is not “forgotten” with time and successive cell generations [30,31].

Epigenetic drivers appear to integrate the sensory and signaling network of endothelial cells, which allows for the control of cardiovascular functions [32]. Endothelial epigenetic operators serve as mechano-transducers in reception, transmission, and integration of atheroprotective and atherogenic hemodynamic forces. Overall, epigenetic alterations play a critical role in arteriosclerosis [16,33,34]. Furthermore, recent doctrines in malignant transformation also include the participation of epigenetic derangements along with genomic mutations, as determinants of canonical carcinogenesis stages as initiation, progression, metastasis, immortality, and autonomy [35]. Phenomena such as histone modifications, nucleosome remodeling, DNA methylation, and miRNA-mediated gene expression control are dynamic writers of the cancer epigenetic code that in concert with the cluster of gene mutations dictate and perpetuate the malignant phenotype [36,37,38].

We have recently investigated the existence of a hypothetical cellular pathologic memory, its interspecies transmissibility from human to rodent, and the imprinting generated in the recipient animal’s tissues. In essence, the consequences of the administration to rodents of simple, crude, cells-free filtrates, derived from human archetypal pathologic tissue samples of type-2 diabetes, arteriosclerosis, and cancer were examined. Here, we describe and discuss in narrative styles the group of these unprecedented findings, and hypothesize on their underlying operators. All the figures and data presented here derive from our studies and are extensively described in the original research articles here discussed.

These studies reinforce the notion on the existence and transmissibility of a pathologic memory encrypted in diseased tissues of humans affected by three different non-communicable diseases. The eventual identification of the drivers and mechanisms operating behind the reproduction in otherwise healthy animals of the histological traits of the pathologic human donor may entail a far-reaching futuristic impact over the control of these diseases.

## 2. Metabolic Memory Is Transferable and Induces the Recapitulation of Histopathologic Hallmarks in Normal Rats

Years ago, we observed that diabetic granulation tissue arterioles somehow “inherit” and recapitulate in a period of a few days, a collection of histopathological hallmarks of chronic evolution that typically characterize diabetic microangiopathy [39]. This observation suggested the existence of an aberrant angiogenesis program in diabetes, which was possibly geared by epigenetic drivers, and that ultimately lined up with the conceptual essence of metabolic memory [40,41]. We, therefore, hypothesized that cells of budding vessel and those recruited endothelial progenitor cells were somehow stamped by a pathologic angiogenesis code, which could be transferred as a soluble signal, and ultimately, impose a diabetic donor’s imprinting on the tissues of a normal host animal. As mentioned, these ideas were undertaken through cell-free homogenates elaborated from fragments of granulation tissue of an ischemic diabetic foot wound, popliteal artery, and peroneal nerve samples—all derived from patients with type 2 diabetes following lower extremity amputations. Post-mammoplasty exuberant granulation tissue and post-traumatic amputation-tissue samples from normal, healthy donors acted as controls of the human-to-rat xenotransplant. All the experiments commented on here used total protein concentration as an arbitrary unit for the material administration dose criteria. Accordingly, different experiments were conducted in which full-thickness wounds were intralesionally infiltrated with cell-free filtrates derived from lower limb ischemic wounds-granulation tissue, non-thrombotic arteriosclerotic popliteal samples, and peroneal nerve. Within seven days of the animal wounds’ exposure to diabetic homogenates, typical histopathological changes of microangiopathy (Figure 1) and neuropathy were generated on the background of healthy recipient rats [42]. In essence, the arterioles of the rats’ wounds mirrored the broad constellation of arterial histopathological changes detected in the soft peripheral tissues of the diabetics’ lower limbs. Considering the crucial pathogenic relevance of advanced glycation end products (AGE) and their receptor (RAGE) interaction in a variety of pathological processes spanning from inflammation and oxidative stress to tumorigenesis, irreversible diabetic complications, and vascular remodeling [43,44,45], we examined the capacity of glycated bovine serum albumin (BSA) to trigger arterial thickening or any other abnormal change. Glycated BSA perturbed the healing trajectory, promoted inflammation, and recreated diabetic neuropathy, but it did not result in arteriolar thickening despite the high concentration of AGE (683.8 ng/mg). We, therefore, share the notion that factor(s) other than acute glucotoxic reactants are factual drivers of the observed vascular remodeling. Immunohistochemistry experiments showed that the recipient animal’s granulation tissue had reproduced the immunoexpression phenotypic pattern of the donor for all the pathogenically relevant markers studied [42].

## 3. Cell-Free Filtrates (CFF) from Non-Diabetic Arteriosclerotic Samples Induce the Recapitulation of Arteriolar Pathology in Host Animals

Encouraged by the hypothetical existence of a “vascular memory” aside from the canonic concept of diabetic’s endothelial memory, we examined the potential pathological significance of administering a cell-free filtrate obtained from non-diabetic amputee patients, affected by critical limb ischemia using the same experimental protocol as for diabetics’ study. It was again observed that in a 7-day administration period, the human arteriosclerotic-derived filtrate induced the recapitulation of angiogenic and arterial anomalies within the host rat-granulation tissue. Interestingly, these rats reproduced the donor’s occlusion pattern characterized by the projection and luminal encroachment of collagen bundles and the presence of fusiform, fibroblast-like cells, apparently replacing endothelial cells [46]. This observation may represent an endothelial-to-mesenchyme reprogramming process [47] (Figure 2). The so-called “aberrant angiogenesis” was also observed, in which muscle myofibrils are replaced by vascular-like channels that may contain an endothelial collar, so that one tissue lineage is substituted by an unrelated one (not shown) [48]. Furthermore, as described for the diabetes study, immunohistochemistry experiments confirmed that the arteriosclerotic material recipient rats entirely recreated the immunoexpression pattern of vascular critical markers found in the pathologic donor samples [46].

## 4. Cell-Free Tumor Tissue Filtrate Transforms Cells and Is Carcinogenic in Nude Mice

A third line of experiments addressed the consequences of injecting healthy nude mice with fresh human tumors cell-free filtrates, assuming the hypothetical transference of a donor’s-derived carcinogenic memory, encrypted into malignant cells and represented as soluble chemical signals. The study included a protocol examining the effects of CFF administration derived from high-grade mammary ductal carcinomas for 6 and 12 consecutive weeks in an increasing protein concentration/dose regimen of the tissue homogenate. A second protocol evaluated the effects of administering a CFF derived from an anaplastic pleomorphic sarcoma at a constant dose of 100 µg of protein for 32 consecutive days. At this time point, half of the animals were destined to an early autopsy (*N* = 6/group), while the other half were left to evolve untreated for other 35 days for a late autopsy (day 68) [49].

Six weeks of treatment with mammary tumor-derived material induced lung parenchymal condensation, consisting of atypical adenomatous hyperplasia along with multifocal nodules of solid adenocarcinomas. Figure 3 shows an exemplary nodule of a solid, poorly differentiated adenocarcinoma, with expression of two well-validated malignancy markers. Mice administered for 12 weeks exhibited a diseased-like behavior, whereas pathology study revealed almost massive parenchymal condensation and alveolar lumen erasing due to invasion of multinodular, solid, and lepidic growth adenocarcinomas. Immunohistochemistry of malignant growth foci showed the overexpression of both EGF and VEGF receptors, CEA, and other transformation markers such as c-Myc, TGF-α, and PCNA (not shown).

Administration of the pleomorphic sarcoma homogenate for 32 days had a far more acute and aggressive course in the mice. The animals developed lung parenchymal consolidation with foci of malignant, poorly differentiated cells, positive to oncofetal markers and concluded as adenocarcinoma (not shown), and a subcutaneous nodule classified as a poorly differentiated mesenchymal cells sarcoma-type tumor (Figure 4).

Animals that evolved treatment-free for other 35 days progressed to a broader incidence and variety of tumors including lung solid adenocarcinomas (Figure 5), multiple foci of epithelial tumors with glandular differentiation within the mediastinal adipose tissue, and an undifferentiated thyroid tumor (not shown). Importantly, none of these alterations were detected in the control group of mice treated with a healthy skin donor-derived homogenate. Conclusively, these experiments showed that it is possible to induce malignant tumors in healthy animals in a short period of time, through the systemic administration of malignant human neoplasias-derived cell-free filtrates.

## 5. Discussion

We have raised the hypothesis that cells contain a disease code memory, that it is simply extractable, passively transferred from human to rodents, and that it ultimately induces the recapitulation in the normal animals’ tissues phenotypic traits evocative of the donor’s pathology. This is a novel and risky hypothesis as it presupposed that a simple cell-free filtrate prepared from tissue fragments derived from non-communicable disease-affected human donors could recapitulate in healthy animals specific attributes that characterize the histology of the donor’s disease. The methodological bases of the experiments discussed here expand on the elaboration and subsequent administration of crude, cell-free filtrates from human pathologic tissue samples with normal sterile saline solution as vehicle with no other chemical processing. This is a simple methodology first inaugurated by Peyton Rous in the early years of the 20th century that have significantly contributed to the advancement of experimental pathology [50]. Although cell-free filtrates have been instrumental for the discovery of oncogenic retroviruses and indirectly of cancer-causing genes, the early methodological approach of Ellermann, Rous, Olson, and others [51,52] does not appear to have been progressively used, nor enriched along the years. Thus, to the best of our knowledge, our studies appear to have been solely preceded by the classics of about a century ago [53]. The fact that our cell-free filtrate involves three examples of chronic, non-communicable diseases, justifies our efforts to discern the actual nature of the acting operators.

All these experiments included concurrent control groups, based on the administration to rats and mice homogenates elaborated from healthy human tissue, mostly during cosmetic surgery. Through these experiments, we investigated whether the pathologic tissue-derived material from chronic NCD diseases could transform the recipient animal tissues’ phenotype, and accordingly somehow recapitulate donor’s histopathological traits. This experimental approach is based on two fundamental hypotheses: (1) mammals’ somatic cells are endowed with the ability of “learning and memorizing” their lifetime stress biography, and (2) cell memory information is encrypted in a chemical code that is soluble and transferable to other animal species via internalization by the recipient’s cells, consequently acting as a priming factor to impose and recreate the donor’s pathologic phenotype.

The first study demonstrated that diabetic tissues contain priming factors that may be extracted from both peripheral (granulation tissue) and internal structures (arteries and nerves), that they are apparently “up-taken” by host animal cells, and that the message derived from these priming factors is able to disrupt the normal angiogenic process and the arterial morphology, ultimately reproducing the histology of microangiopathy. Although the mechanisms underlying diabetic metabolic memory and vascular complications remain to be fully elucidated [20,54], this study, in addition to offering an unprecedented system to recreate histological features of diabetic ulcers in a non-glucotoxic environment, incites to consider that diabetic tissue damages can be induced by soluble circulating messengers, and that vessels and nerve damages and other complications could possibly evolve in shorter periods of time as it is contemporarily considered [19,55].

We believe that these messengers are of an epigenetic nature and belong to a large diabetic secretome, which contributes to perpetuating multi-organ complications, and are an active internal force toward cells senescence and ulcer recurrence [56,57].

Having observed that beyond glucotoxicity, other unidentified factors seem to be instrumental for the acute onset of arteriolar thickening and a group of angiogenesis defects, our second study addressed the consequences of the intralesional infiltration of arterial tissue-derived homogenate from non-diabetic subjects, amputated due to chronic limb ischemia. The experiments reproducibly showed the recapitulation of human donor’s arteriosclerosis within the rats’ granulation tissue arterioles in a period of days. Of note, rats’ arterioles faithfully recreated the luminal occlusion and the fibroblast-like cells encroachment, apparently secondary to reprogramming events of the endothelial collar, as it was observed in the donors’ tissues. Since these endothelial cells’ phenotypic transformation is entitled as an endothelial-to-mesenchymal transition (EndMT) process [47], the hypothetical possibility exists that the transdifferentiation drivers involved were passively transferred from the human arteriosclerotic vessel to the normal rat. Epigenetics promoters of EndMT events and members of the transforming growth factor-β (TGF-β) family are invoked as causal factors of the endothelial transdifferentiation process [58,59]. In support of the notion that the pathologic human-derived homogenate locally interfered with the rats’ tissues differentiation homeostasis, is the recapitulation of the “aberrant angiogenesis” within the rats skeletal muscle fibrils, which also represents a transdifferentiation episode [48]. This study contributes to nurturing the hypothesis of an existing “pathologic vascular memory” that surpasses the limits of the legendary “vascular glycemic memory”. It was evidenced that non-diabetic diseased arteries include a surge of “priming factors” that appear to be soluble, can be transferred, and impose the “damage phenotype” in naïve recipient tissues [37]. Accordingly, and in terms of diabetes, we deem that these “vascular memory factors” are circulating messengers that may contribute to arterial disease debut, perpetuation, dissemination, and irreversibility [46].

Common for the two studies discussed so far is the fact that immunohistochemistry experiments showed that animals receiving the pathologic tissue-filtrates authentically mirrored the immunoexpression pattern of the donor samples, in sharp contrast with control animal groups. This includes over and sub-expression of biomarkers related to inflammation, as well as vascular physiology and pathology. We consequently hypothesize that the recapitulation of the donor’s morphological traits in rats’ arterioles is the consequence of the paracrine expression and production of pathologic vascular remodeling messengers derived from the diseased human tissues. These vascular “remodelome” may entail transcription factors, fragments of abnormally methylated DNA sequences, microRNAs, and other epigenetic operators.

The fact that none of the described vascular changes were detected in animals treated with healthy donor tissue homogenates supports the authenticity of the findings, and the possibility that these alterations do not correspond to the host’s tissue reactivity to human xenogeneic material. Common to these two studies is the interspecies transmission of the pathologic code irrespective of the specific disease of the donors. We currently ignore the actual nature of the drivers operating behind the recapitulation of the human arterial pathological process in the otherwise normal rats. However, we think that these recapitulation events are promoted by some form of horizontal interspecies genetic material transmission that may include epigenetic drivers contained in the broad constellation of cellular constituents within the cell-free filtrates used here.

Our third study discussed here [49] is, to our knowledge, the first demonstration of malignant tumor development in otherwise healthy mice by the administration of simple, aqueous homogenates derived from non-transmissible human tumors. Of note, and as noted for the two previous studies here described, these pre-malignant and malignant lesions were uniquely induced by the administration of the whole-tissue homogenates using sterile physiologic saline solution. Consequently, these homogenates represent a rich-in-content material and a vehicle for donor cells’ pathologic signatures. Overall, this study suggests that malignant tissues are also endowed with a “malignant code” that can be passively transferred to other host mammal cells, imposing its legacy, and ultimately implementing a carcinogenesis process in vivo. According to our experimental pathology data, these transferred “malignant signalers” not only disrupted cells proliferation control, but also altered the differentiation program of normal, mature populations of epithelial and mesenchymal cells in mice organs. The fact that the initial areas of alveolar atypical adenomatous hyperplasia progressed to authentic, invasive, and metastatic tumors in the case of lung adenocarcinomas induced by breast tumor homogenate, indicate the transit through the lineal transformation phases of “initiation–promotion–progression”. All these events evolved in short temporary windows, which leads to questions about what drivers and forces could be propelling the rapid tumor growth. One of the most remarkable demonstrations here is that when mice were treated with the sarcoma homogenate for a month and subsequently left inducer-free for other 35 days, there was an increase in the number and variety of tumors and affected organs. This observation is compatible with Hanahan and Weinberg’s tumors hallmarks that include tumor cells self-capability, irreversibility, and autonomous growth [60]. We, therefore, hypothesize that these emblematic malignancy traits are genetically and/or epigenetically driven, and consequent to some type of eukaryotic-to-eukaryotic horizontal transfer of genetic and/or epigenetic effectors. This notion is somewhat supported by the presence of integral DNA and RNA molecules in the tumor cell-free filtrates despite the mechanical processing of tissue disruption. Previous studies demonstrate that nucleic acids primarily packed into exosome vesicles play a seminal role in most of the carcinogenesis stages [61,62]. Our tissue homogenates may also contain some type of tumor or other pathologic tissue-derived extracellular vesicles, such as exosomes [63], which are exemplary players in cell–cell communication by conveying genetic and epigenetic messages [64]. The pathogenic engagement of exosomes in human cancer is compelling given their ability to deliver DNA and RNA-tumorigenic signatures that participate in malignant transformation [65,66]. To summarize, two different classes of human malignant tumors-derived cell-free filtrates were proven to transform mice cells and induce a carcinogenic effect. This neoplastic growth may reflect the influence exerted by putative “malignancy messengers” contained in the human pathologic material that incorporate into animal’s cells, and dominantly impose the donor’s abnormal phenotype over the host’s permissive environment.

We have discussed here evidence of three in vivo studies based on the administration of a homogenate from human pathologic tissue samples of type-2 diabetes, arteriosclerosis, and cancer as illustrative of expanding, chronic, non-communicable diseases. Although we have not elucidated the mechanisms behind these interesting findings, the replications in normal animals of histopathological changes typical of the human diseased tissues support our hypothesis on the existence of a transmissible “epigenetic cellular disease memory”. In line with this, mounting data continue to support the idea that behind an intrauterine harassment and the ensued inheritance of a pathologic trait, it is the protagonist role of epigenetic mechanisms [67] that, under de novo rewritten abnormal signatures, represent the epigenetic pathologic memory [68]. This memory ensures the transgenerational transmission and phenotypic expression of the consequences of the suboptimal intrauterine environments [69], providing the basis for the “developmental origins of health and disease” concept [68]. In agreement with the pathogenic role of metabolic memory for diabetic complications onset and progression are the toxic consequences of the intrauterine-shaped epigenetic memory by hyperglycemia, as a major driver of a constellation of successive transgenerational alterations in descendants [68,70,71,72]. Unrelated to glucose metabolism stands the experimental demonstration that in utero exposure to nicotine imposes permanent epigenetic programs that are transferred through the germline to subsequent generations, leading to asthma predisposition [73]. The participation of epigenetic memory has also been invoked in the transgenerational transmission of behavioral alterations following in utero immune activation, so that prenatal immune challenge may impair cerebral activity across multiple generations [74]. In a similar manner, the transgenerational inheritance of posttraumatic stress disorder effects has been documented in humans, which ultimately emphasizes the critical role of epigenetic mechanisms in the complex interface of environment and biology [75]. In conclusion, we deem that the disease-related epigenetic fingerprints that make up the fundamentals of the “pathologic memory” are capable of intraorganismal and transgenerational transmission, which leads us to reconsider the concept of non-communicable diseases for those processes in which there is not a causal living pathogen. Indeed, there is a long and winding road ahead.

We deem these studies to offer a useful experimental platform for translational medical research for non-communicable diseases. The ultimate identification and pharmacologic validation of these driving pathogenic players may reveal future preventive and therapeutic alternatives.

## Figures and Tables

**Figure 1 ijms-23-11575-f001:**
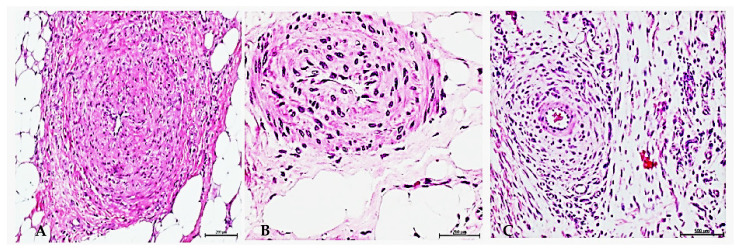
Rats’ arteriolar thickening after exposure to diabetic tissue cell-free filtrate. (**A**) Image showing luminal collapse of arterioles derived from the granulation tissue of a diabetic donor exhibiting intima hyperplasia, media layer thickening, hypercellularity, and concentric collagen of hyaline aspect. H/E. Magnification ×20. (**B**) Rat’s arteriolar walls thickening with luminal collapse in which intima and media layers appear hyperplastic with hypercellularity and fused as a result of infiltrated diabetic tissues homogenate. H/E. Magnification ×40. (**C**) Concentric adventitial collagen accumulation and hypercellularity around an arteriole in a rat treated with diabetic tissues homogenate. H/E. Magnification ×20.

**Figure 2 ijms-23-11575-f002:**
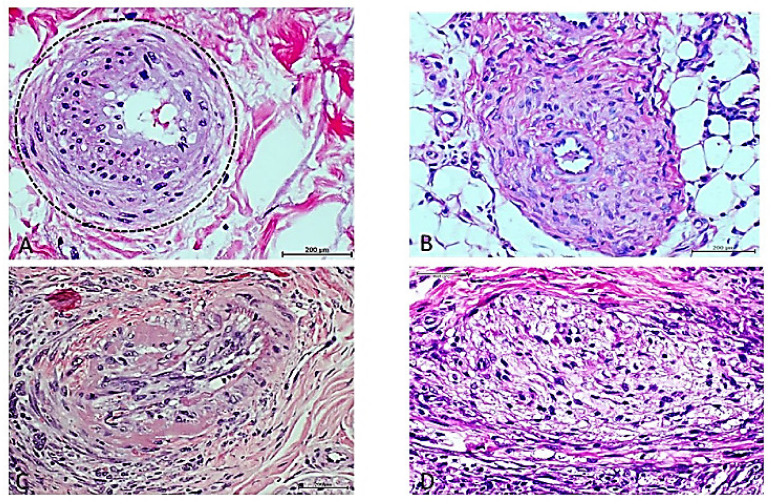
Rats’ arteriolar remodeling after exposure to non-diabetic arteriosclerotic arterial tissue. (**A**) Human donor arteriole with an active process of intimal hyperplasia, subendothelial infiltration, luminal narrowing and concentric expansion of the media layer with large basophilic nuclei. H/E. Magnification ×40. (**B**) Vascular response to human arteriosclerotic material in which a rat arteriole exhibits wall thickening, abnormal cellular infiltration, and disorganization. There is an exaggerated accumulation of collagen bundles. The lumen is obliterated. H/E. Magnification ×40. (**C**) Human donor arteriole showing degenerative wall changes including accumulation of hyaline material and internal tunica elastica fragmentation. Endothelial cells are hypertrophied and intermixed with fibroblast-like cells that appear to emerge from the endothelial collar and make up a mesh-like structure that encroaches into the lumen. H/E. Magnification ×40. (**D**) Rat’s arteriole with luminal invasion by protruding fusiform-like cells that emerge from the endothelial side and project into the lumen, forming a trabecular structure. The luminal mesh is infiltrated by round basophilic nuclei, suggestive of lymphocytes and other of fibroblastic aspect. H/E. Magnification: ×40. Scale bar: 200 µm.

**Figure 3 ijms-23-11575-f003:**
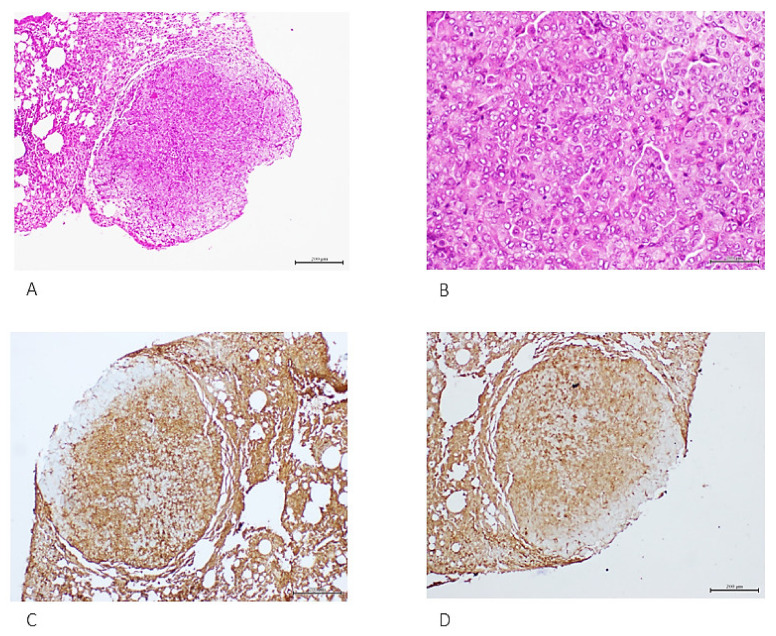
Lung solid tumor nodule detected at the early autopsy time point. (**A**). Well-consolidated tumor nodule with parietal, sub-pleural growth. The histological analysis identified poorly differentiated epithelial cells with vacuolated cytoplasm, vesiculous nuclei and prominent nucleoli, (**B**). This tumor was concluded as a poorly differentiated adenocarcinoma reactive to CEA and TTF-1, (**C**,**D**) respectively. H/E. Magnifications: ×4 and ×40. Scale bar: 200 µm.

**Figure 4 ijms-23-11575-f004:**
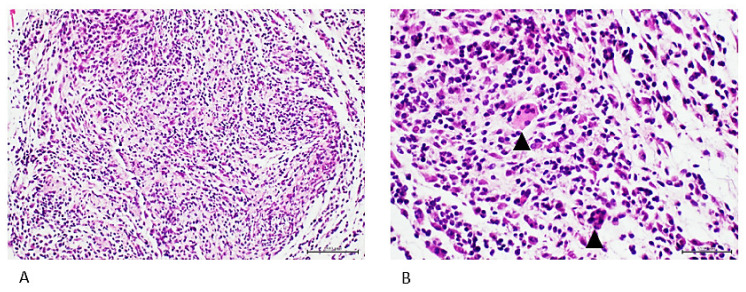
The mesenchymal cells tumor. This was a readily detected interscapular nodule during necropsy. (**A**) Panoramic image (×4) demonstrating intense cellularity and local disorganization. (**B**) Larger magnification (×40) reveals intense basophilic and pleomorphic nuclei, cells with embryonic aspect and giant multinucleated cells (arrowhead). H/E. Scale bar: 200 µm.

**Figure 5 ijms-23-11575-f005:**
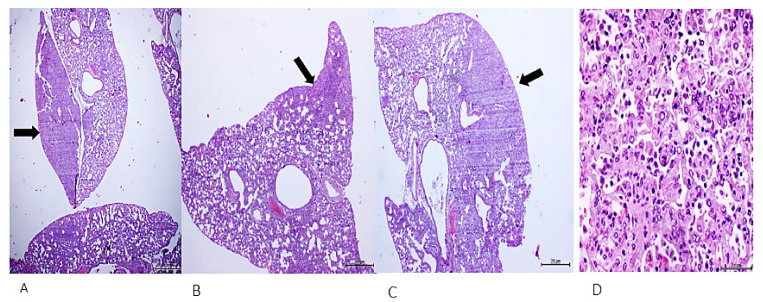
Autonomous growth of lung tumors. Mice treated with the sarcoma homogenate for 32 days and left to evolve free-of-treatment for other 35 days, exhibited lungs parenchymal consolidated areas (**A**–**C**) that were histologically concluded as poorly-differentiated adenocarcinomas. H/E. Magnification: ×4 from (**A**–**C**) and ×40 for (**D**). Scale bar: 200 µm.

## Data Availability

This is a narrative review based on three studies developed and published earlier by our group. We are delighted to share and discuss data presented in the articles and would make available other unpublished evidences.

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
