# Peer review of "Cell-Free Filtrates (CFF) as Vectors of a Transmissible Pathologic Tissue Memory Code: A Hypothetical and Narrative Review"

_ijms, 2022, doi:10.3390/ijms231911575_

Round 1

Reviewer 1 Report

The Manuscript entitled "Cell-free filtrates as vectors of a transmissible pathologic tissue 2 memory code. A hypothetical and narrative review" by Jorge Berlanga-Acosta et al., has merit in the present hypothesis. Authors have discussed about the cellular memory representing cells ability to “write and memorize” stressful experiences via epigenetic operators. Cultured diabetic and cancer patients-derived cells recall behavioral traits as when in the donor’s organism irrespective to culture time and conditions. Authors have analyzed the data of studies conducted by them and led by a cascade of hypothesis, in which they address to validate the hypothetical existence and transmissibility of a cellular pathologic memory in diabetes, arteriosclerotic peripheral arterial disease, and cancer. The administration of each pathologic tissue homogenate consistently induced the faithful recapitulation of: (1) Diabetic archetypical changes in cutaneous arterioles and nerves. (2) Non-thrombotic arteriosclerotic thickening, collagenous arterial encroachment, aberrant angiogenesis, and vascular remodeling. (3) Pre-malignant and malignant epithelial and mesenchymal tumors in different organs; all evocative of the donor’s tissue histopathology and with no barriers for interspecies transmission. Author has hypothesize that homogenates contain pathologic tissue memory codes represented in soluble drivers that “infiltrates” host’s animal cells, and ultimately impose their phenotypic signatures. The identification and validation of the actors in behind may pave the way for future therapies. The hypothesis is good, and may be considered, provided authors provide suitable rebuttal to the queries raised-

Comment 1. How did the authors infer that this hypothesis is new and novel?

comment 2. Authors have discussed about the AGE and RAGE in just a line without giving the background of AGE and RAGE. They are encouarged to give the little background in 3-4 lines with proper citations like,

i. Do all roads lead to the Rome? The glycation perspective!. Seminars in Cancer Biology 2018: 49, 9-19.

ii.  An Immunohistochemical Analysis to Validate the Rationale behind the Enhanced Immunogenicity of D-Ribosylated Low Density Lipo-Protein. PLoS ONE. 2014;9(11).

iii. The receptor for advanced glycation end products: A fuel to pancreatic cancer. Seminars in Cancer Biology 2018: 49, 37-43.

comment 3. Authors are advised to check for english, grammar error throughout the paper.

comment 4. As it is the hypothesis paper therefore if the figures used by the authors is taken from other source should have proper permission.

comment 5. The figures used here from the authors have no methodology described, if the figures/data is generated from their own lab it should be specifically defined in the hypothesis.

Author Response

Dear Reviewer 1. Thank you very much for the excellent and professional examination to our mansucript. We have addressed each and every comment. Please find attached our responses. Thank you very much. 

Reviewer 2 Report

Berlanga-Acosta and collaborators present and discuss previously published evidence that pathologic human tissues contain bioactive molecules able to induce in recipient healthy rodents phenotypic traits of the donor's pathology. The data obtained in their laboratories with tissue cell-free homogenates from subjects with various diseases (type 2 diabetes, arteriosclerosis, and cancer) are discussed in light of the existing literature, leading the authors to the formulation of a new hypothesis, i.e. that pathologic tissues contain a memory code transmissible to healthy tissues of recipients of a different species. Since they administered to animals pathologic cell-free filtrates they suggest that such a code is likely constituted by metabolites or epigenetic modifiers able to superimpose their effects on normal cells.

The hypothesis is interesting and deserves further research to characterise the chemical nature of the pathologic secretome. I would encourage the authors to add to the manuscript any preliminary data they might have or, alternatively, to discuss and comment work of others that attempted to characterise pathologic cell-free filtrates applied in vitro to cultured cells or organoids obtaining similar results to those here presented. It is possible that also portions of cell membranes were present in their cell-free filtrates as exosomes may not have been retained by the filter they used (please specify the filter type and discuss also this aspect).

Another question arising from this hypothesis is that a pathologic memory code may be transmissed to the progeny of affected subjects. Indeed, offsprings of women with hypertension and diabetes during pregnancy for instance have been reported to be at  higher risk of such diseases. Clearly - in this case - genetic factors need to be dissected from metabolic factors.  Perhaps authors may wish to include their thoughts on gene-independent disease transmission to offsprings as well.

Overall, an intriguing picture emerges, that is non communicable diseases may actually be ... communicable!

Minor points:

Page 3, Figure 1: panels need to be labeled.

Page 4, lines 159-162: comments from a previous reviewer need to be removed.

Page 5, line 183: CFF needs to be defined at first appearance in text.

Page 10, line 353: Author contribution is missing.

Page 10, References: last page of cited articles is not always present. Please check entire list of references.

Author Response

Dear Reviewer 2. Thank you very much for your excellent and professional examination to our manuscript. Thank you for your notes and ideas. Thank you for understanding our hypothesis. Please find attached our responses to each detail. Thank you very much. 

Round 2

Reviewer 1 Report

No further comments.